# Colonic Bacteria-Transformed Catechin Metabolite Response to Cytokine Production by Human Peripheral Blood Mononuclear Cells

**DOI:** 10.3390/biom9120830

**Published:** 2019-12-05

**Authors:** Rajapandiyan Krishnamoorthy, Abdulraheem R. Adisa, Vaiyapuri Subbarayan Periasamy, Jegan Athinarayanan, Subash-Babu Pandurangan, Ali A. Alshatwi

**Affiliations:** Nanobiotechnology and Molecular Biology Research Lab, Department of Food Science and Nutrition, College of Food and Agriculture Sciences, King Saud University, Riyadh 11541, Saudi Arabia; rkrishnamoorthy@ksu.edu.sa (R.K.); vsperrys@gmail.com (V.S.P.); jegan.dna@gmail.com (J.A.); subashbabu80@gmail.com (S.-B.P.)

**Keywords:** anti-inflammatory, cytokine, immunomodulation, human peripheral blood mononuclear cells, catechin, metabolites, proliferation, antioxidants

## Abstract

Human gut microbes are a profitable tool for the modification of food compounds into biologically active metabolites. The biological properties of catechins have been extensively investigated. However, the bioavailability of catechin in human blood plasma is very low. This study aimed to determine the biotransformed catechin metabolites and their bioactive potentials for modulating the immune response of human peripheral blood mononuclear cells (PBMCs). Biotransformation of catechin was carried out using in-vitro gut microbial biotransformation method, the transformed metabolites were identified and confirmed by gas chromatography-mass spectrometry (GC–MS) and high-performance liquid chromatography-mass spectrometry (HPLC–MS). Present observations confirmed that the catechin was biotransformed into 11 metabolites upon microbial dehydroxylation and C ring cleavage. Further, immunomodulatory potential of catechin metabolites was analyzed in peripheral blood mononuclear cells (PBMCs). We found up-regulation of anti-inflammatory cytokine (IL-4, IL-10) and down-regulation of pro-inflammatory (IL-16, IL-12B) cytokine may be due to Th2 immune response. In conclusion, biotransformed catechin metabolites enhance anti-inflammatory cytokines which is beneficial for overcoming inflammatory disorders.

## 1. Introduction

Human health is directly influenced by the immune system, which is designed to provide the host protection against harmful foreign substances. Almost all infectious diseases are the result of an inadequate immune response [1]. Therefore, the modulation of the immune system is of great importance for the control of many immunological disorders. Green tea has been consumed for centuries due to its health benefits [2]. Green tea extract contains different polyphenols, such as catechin (C), epicatechin (EC), epigallocatechin gallate (EGCG), and epicatechin gallate (ECG), that are responsible for various health benefits, such as antioxidant, antibacterial, and anticancer properties, and protection against cardiovascular diseases, among others [3]. A number of studies have demonstrated that catechin is poorly absorbed and is excreted to a small extent in urine [4,5]. Subsequently, the fate and bioavailability of catechin, including intestinal absorption, distribution in human tissue, and excretion, must be established.

The gut microbiota transforms the phytocompounds into numerous bioactive metabolites. Recent studies have suggested that catechin can be degraded and/or modified by many intestinal bacteria [6], thus significantly altering the parent compound [7]. In recent decades, studies have focused on catechin biotransformation by the microorganisms present in the human gut. Despite this exponential increase in microbiome research, the link between biotransformation and the role of biotransformed metabolites still remains largely unexplored. Recently, reports have suggested that biotransformed catechin metabolites exert several biological activities, such as the inhibition of platelet aggregation and the activation function, the inhibition of cyclooxgenous-2 in colon cancer cells, and antiproliferative activity in prostate and cancer cells [8,9].

Low-grade chronic inflammatory processes result from the interaction between monocytes, T lymphocytes, endothelial cells, and smooth muscle cells [10]. Predominantly, subsets of monocytes and T lymphocytes play a critical role in immunoregulation. Several studies have highlighted the immunomodulatory effects of biotransformed or bio-active metabolites on proinflammatory and anti-inflammatory cytokine production in different in vivo and in vitro models. Several studies have reported that polyphenol-related gut bio-active metabolites modulate the immune response through the stimulation of certain anti-inflammatory cytokines and decreased proinflammatory cytokine production [11,12,13]. The human gastrointestinal tract contains high level of biotransformed metabolites, which can penetrate host tissues and interact with immune cells, thereby triggering host immunological response [14]. However, no reports are available on the biotransformed catechin and how it responds to anti-inflammatory cytokines (Interleukins-IL-1A, IL-1B, IL-4, IL-5, IL-6, IL-10, IL-13, transforming growth factor beta (TGF-β) and pro-inflammatory cytokines (IL-2, IL-12, IL-17A, interferon gamma (INFγ), granulocyte-macrophage colony-stimulating factor (GM-CSF), tumor necrosis factor alpha (TNF-α).

The gut microbe produces a larger toolkit of enzymes that catalyze the diverse range of chemical reactions and results in transformation of catechin. However, this alteration can cause increase or decrease the activity on human health [15]. The aim of the present study was to investigate in vitro microbial biotransformation and the effect of biotransformed metabolites on the proliferation of human peripheral blood mononuclear cells (PBMCs), specifically the production of pro-inflammatory cytokines and anti-inflammatory cytokine profiles such as IL-2, IL-4, IL-5, IL-6, IL-10, IL-12, IL-13, IL-17A, interferon gamma (IFN-γ), transforming growth factor-β, (TGF-β), tumor necrosis factor-β (TNF-β), and GM-CSF. The expression of genes related to human PBMCs was also studied.

## 2. Materials and Methods

### 2.1. Reagents

3-(4,5-dimethylthiazol-2-yl)-2,5-diphenyltetrazolium bromide (MTT) and Histopaque were purchased from Sigma (St. Louis, MO, USA) and RPMI-1640 medium was obtained from Invitrogen (Grand Island, NY, USA). Multianalyte ELISA array kits for 12 cytokines were purchased from Qiagen (MEH004A, Qiagen, Hilden, Germany). Distilled water was obtained using a Milli-Q system (Millipore Laboratory, Bedford, MA, USA). Deionized water was obtained using a Direct-QUV 3 Multipore Water purification system (Millipore, Burlington, MA, USA). All other chemicals and reagents were of analytical grade unless otherwise specified.

### 2.2. Preparation of Fecal Slurry

Fecal samples were obtained from three healthy volunteers who had usually taken a normal diet and had not received any antibiotics for a minimum of six months prior to stool collection. All subjects gave their informed consent for inclusion before they participated in the study. The study was conducted in accordance with the Declaration of Helsinki, and the protocol was approved by the Institutional Review Board of King Saud University College of Medicine (E-19-3703). Fecal bacterial suspensions were prepared by the method described previously [16]. Briefly, the collected fresh samples were stored in 4 °C in to an anaerobic chamber and diluted (1:10 *w*/*v*) with 0.1 M phosphate buffer (pH 7.2) within 1–2 h and mixed uniformly by vortexing. The resultant fecal slurries were pooled, slurry was centrifuged at 4000× *g* for 10 min, and resultant fecal bacterial suspension inoculated into fermentation vessels containing minimal medium.

### 2.3. Fecal Batch-Culture Fermentation

Fecal batch-culture fermentation was carried out as previously reported [17]. Briefly, five separate 500-mL glass fermenter vessels were filled with 250 mL of sterilized medium (peptone bacteriological (2 g/L), NaCl (0.1 g/L), yeast extract (2 g/L), K_2_HPO_4_ (0.04 g/L), KH_2_PO_4_ (0.04 g/L), resazurin (1 mg/L), MgSO_4_·7H_2_O (0.01 g/L), NaHCO_3_ (2 g/L), CaCl_2_·6H_2_O (0.01 g/L), hemin (50 mg/L), Tween 80 (2 mL/L), vitamin K (10 μL/L), bile salts (0.5 g/L), l-cysteine (0.5 g/L), and double-distilled water. The medium was regulated to a pH of 7.0 and constantly sparged with O_2_-free N_2_ overnight. The pH was maintained at 7.0 and the temperature at 37 °C, so as to simulate the conditions of the human gut. The vessels were inoculated separately with 15 mL of fecal slurry (1.5% *w*/*v*) along with 100 mM catechin (C0567, Sigma-Aldrich). A negative control was maintained in a separate batch-culture vessel prepared under the same conditions without catechin. Anaerobic conditions were maintained and biotransformed metabolites harvested at 24 h, then immediately stored at −80 °C for further analysis.

### 2.4. Separation and Purification of Biotransformed Metabolites

The biotransformed metabolites were separated by the method described in Takagaki and Nanjo [18] with some modifications. The entire crude culture samples were centrifuged at 7000× *g* for 10 min at 10 °C. The resulting supernatants were collected and filtered using a 0.25-µm Millipore microfilter. The collected filtrates were extracted three times with an equivalent volume of ethyl acetate. Excess solvents from biotransformed crude extract (BCE) were removed by roto-evaporator under controlled temperature (40 °C). Further, the BCE were subjected to preparative thin layer chromatography using mobile phase 0.5% (*v*/*v*) acetic acid in 60% (*v*/*v*) aqueous methanol. Two visible spots were observed, these procedures were repeated ≈ 50 times and the respective spots were pooled separately in order to obtain the compounds quantitatively. The separated compounds were named as biotransformed metabolites A (BTMFA) and biotransformed metabolites B (BTMFB), and all were pooled, filtered using a PVDF 0.22-µm microfilter and further profiled by gas chromatography and mass spectrometry (GC–MS) and high-performance liquid chromatography/mass spectrometry (HPLC–MS).

### 2.5. GC–MS Analysis

The pooled fecal samples, control (fecal sample along with medium without catechin) and separated biotransformed metabolites were analyzed by GC–MS using the modified method [19] with a ZB-5MS 30 m × 0.25 i.d. × 0.25 μm capillary column (Phenomenex, Cheshire, UK) with an injection volume of 10 μL, an initial temperature of 70 °C for 10 min, which was increased to 150 °C at 10 °C/min for 15 min and then to 220 °C at 5 °C/min for 15 min, an injector temperature of 270 °C, an MS transmission line of 280 °C, an ion source at 190 °C, and a split ratio of 1:100. Scanning of mass was carried out at 50–650 *m*/*z* at 0.82 scans/s. The electron impact energy was 70 eV. Biotransformed compounds were identified in relation to their retention time, with the mass spectra of authentic standards from the NIST 98 library.

### 2.6. HPLC-MS Analysis

HPLC were performed using Agilent 1290 infinity system (Agilent Technologies Inc, Waldbronn, Gremany) coupled with quadrupole LC/MS Agilent 6120 (Agilent Technologies Inc). The samples were injected on to a C-18 column (4.6 25 cm, 5l m; phenomenex, Torrance, CA, USA). The solvent used were A—90% acetic acid–water and B—10% MeOH, establishing following elution gradient; isocratic 10% B for 5 min, 10–100% B over 10 min, 100% B for 6 min, and re-equilibration of the column using flow rate of 0.1 mL/min. The spectra were recorded in negative and positive ionization mode between *m*/*z* 50 and 1200.

### 2.7. Immunomodulatory Activity of Biotransformed Metabolites on PBMCs

One hundred milliliters (100 mL) of venous blood was collected from a single healthy donor who had not consumed any antioxidant-containing foods (e.g., salads, fruits, and natural/manufactured juices) 24 h prior to blood collection. PBMCs were isolated using Histopaque (Histopaque-1077, Sigma) density gradient methods by centrifuging for 15 min at 2500× *g* [20] with some modifications. The concentration of PBMCs was adjusted to 1 × 10^4^ cells/mL in RPMI-1640 culture medium. The cells were maintained in RPMI-1640 culture medium supplemented with 10% FBS at 37 °C and 5% CO_2_ in a humid environment for 24 h.

### 2.8. Cytotoxicity (MTT) Assay

The MTT assay was used to examine cell viability. Briefly, the isolated primary PBMCs (lymphocytes, monocytes) were seeded at a density of 1 × 10^4^ cells/mL in 96-well plates. The pooled biotransformed metabolites, BTMFA and BTMFB, and BCE were treated with different doses (0, 20, 40, 60, 80 and 100 µg/mL). After 24 and 48 h incubation, 20 µL (MTT, 5 mg/mL) of 3-(4,5-dimethyl-2-thiazolyl)-2,5-diphenyl-2H-tetrazolium bromide were added to each well. The plates were incubated for 4 h to overnight, and the supernatant was then removed by centrifugation. The insoluble formazan was dissolved with dimethyl sulfoxide (DMSO 100%) and the optical density was measured at 570 nm and reference filter (630 nm) using a microplate reader (Promega, Madison, WI, USA). From the values obtained, the percentage viability (relative to survival of control cells) was calculated. Each assay was performed three times (in duplicate/dose) and data are presented as the percentage mean *±* SD.

### 2.9. Cytokines Analysis

BTMFA, BTMFB, and BCE were tested for their effect on the production of pro-inflammatory and anti-inflammatory cytokines in unstimulated PBMCs. The metabolites were dissolved in DMSO and further diluted in RPMI-1640 cell culture medium. The final concentration of DMSO in the cells was less than 0.1%. In the control experiments, this concentration did not show any effects on the measured parameters. PBMCs were added (1 × 10^4^ cells/mL) to 96-well plates and pre-incubated for 1 h at 37 °C in a humidified environment containing 5% CO_2_. After treatment, the supernatants were collected after centrifugation and stored at −80 °C until further analysis. The level of cytokines was determined by using two multi-analyte ELISArray kits (Qiagen, Hilden, Germany) (pro-inflammatory cytokines IL-2, IL-12, IFN-γ, GM-CSF, and TNF-α, and anti-inflammatory cytokines IL-4, IL-5, IL-10, IL-13, IL-17A, and TGF-β1) according to the manufacturer’s protocol. Briefly, 50 µL of assay buffer were added to each well of ELISArray plate and then 50 µl collected test samples and control samples were added to appropriate well and incubated for 2 h at 37 °C. The unbounded cytokines were removed and 100 µL of detection antibody solution were added to each well and incubated for 1 h and wells were washed three times with washing solution and 100 µL of Avidin-HRP (Qiagen) was added to each well and incubated for 30 min. After washing, developmental followed by stop solution were added to each well at respective intervals. Finally, the level of cytokines was detected at 450 nm using a microplate reader (Promega).

### 2.10. Quantitative Reverse Transcription-Polymerase Chain Reaction (qRT-PCR)

Complementary DNA (cDNA) was synthesized directly from treated and untreated human WBC using FastLane Cell cDNA Kit (Qiagen, Hilden, Germany). The purity was checked and quantified using the NanoDrop spectrophotometer ND-1000 (NanoDrop Technologies Inc., Wilmington, DE, USA). Quantitative real-time PCR was performed with the SYBR Green PCR Master Mix (Qiagen, Chatsworth, CA, USA) in an ABI 7500 fast real-time PCR system (Applied Biosystems, Foster City, CA, USA) according to the manufacturer’s protocol. The mRNA levels of IL-4, IL-10, IL-12B, IL-16, and the reference gene, glyceraldehyde 3-phosphate dehydrogenase (GAPDH), were assayed using gene-specific SYBR Green-based QuantiTect^®^ Primer Assays (Qiagen, Hilden, Germany) according to manufacturer’s instructions. A 25-μL reaction volume was used in each well of the PCR plates. Briefly, 12.5 μL of master mix, 2 μL of assay primer and 2 μL of template cDNA (500 pg) were added to each well. Reaction mixtures were incubated for initial denaturation at 95 °C for 5 min, followed by 40 PCR cycles. Each cycle consisted of 95 °C for 5 s and 60 °C for 30 s. The relative quantification of the aforementioned gene expression was analyzed using an ABI 7500 Fast Real-Time PCR system (Applied Biosystems). The values were expressed as fold changes over the control and expressed as means with their standard errors.

### 2.11. Statistical Analysis

Data were analyzed using ANOVA (SPSS/11.5 software package, IBM Corporation, Armonk, CA, USA) followed by Tukey’s test. All data are expressed as mean ± standard error of the mean (SEM). Differences were considered to be significant if the *p*-value was ≤ 0.05.

## 3. Results and Discussion

### 3.1. Profiling of Biotransformed Metabolites

The metabolites present in BTMFA and BTMFB were profiled by GC/MS (Figure 1 and Figure 2). A total of 11 (10 metabolites from BTMFA and 1 metabolite from BTMFB) biotransformed catechin metabolites were identified. BTMFA consisted of the following simple form phenolic metabolites: dehydroquinic acid (1), 4-ethylphenol (2), 4-methoxyphenyl propan-2-ol (3), 3-phenyl propionic acid (4), 2-phenoxyethanol (5), benzene tricarboxylic acid,1,2-dimethyl ester (6), catechol-1,4-benzenediol (7), benzene-1,3,5-tris1-methylpropyl (8), 3,5,7-trihydroxy-2H-chromen-2-one (9), and 4-hydroxyphenylpropionic acid (10). BTMFB were rich in dimethoxycinnamic acid (B1). The presence of all the metabolites were confirmed by HPLC-MS analysis based on m/z. The obtained data were listed in the Appendix A. None of the metabolites/ precursor were found in the control GC/MS analysis (Appendix A). The literature confirms that the compounds obtained were phenolic metabolites and are beneficial to human health [21,22].

Gas chromatography–mass spectrophotometry (GC–MS) profiles of catechin biotransformed metabolites by human gut microbes. 

The catechin structure has the potential to strongly interact with bacterial enzymes due to its benzenoid ring and the hydrogen bonding potential of its hydroxyl groups [23,24]. Several isoflavone metabolizing gut bacteria such as *Eggerthella lenta, Flavonifractor plautii*, *Adlercreutzia equolifaciens, Asaccharobacter celatus, Slackia equolifaciens, and Slackia isoflavoniconvertensare* are capable of transforming catechin in the intestinal tract as a result of dehydroxylation and C-ring cleavage to form metabolites 7, 10 and B1 [6]. *Lactobacillus* is capable of reducing catechin by inducible enzymes under in vitro anaerobic conditions to produce metabolites 1, 4, 6, and B1 [25]. The metabolites 2, 3, 4, 7, 10 and B1 were identified as the major catechin metabolites in human blood plasma and urine samples collected from participants 6–48 h after drinking tea [19]. The consumption of catechins by humans results in an increase of metabolite 4 in the blood plasma [26] and an increased urine excretion of metabolites 1, 3, 4, 10 and B1 [27,28]. In vivo animal studies have shown similar results in terms of urinary metabolites from rats after feeding with a catechin diet [29]. These metabolites were also found in an in vitro experiment as a major product of colonic metabolism of polyphenols by human fecal microbiota [30]. 

### 3.2. Effect of Biotransformed Catechin Metabolites on the Cell Population of Human PBMCs

A number of studies on the role of polyphenols in PBMC proliferation, differentiation, and activation for the regulation and determination of immune function are available [21]. In recent years, research on the biotransformation of polyphenols has shed some light on biotransformed polyphenols in response to immune function. In the present study we evaluated the immunomodulatory effect of biotransformed catechin metabolites. BTMFA, BTMFB, and BCE from catechin and their effects on the cell population of PBMCs (monocytes, T lymphocytes, and B lymphocytes) were investigated by MTT analysis. We measured the viability of cell culture medium alone and biotransformed metabolites at varying concentrations (20, 40, 60, 80, and 100 µg/mL). PBMC viability in culture medium alone was in the range of 99–100%, BTMFA treatment in the range of 90–100% viability (Figure 3(A1–A3)), and BTMFB treated in the range of 84–86% viability after 48 h (Figure 3(B1–B3)). Pure catechin treatment with PBMC revealed the decreasing cell viability of unstimulated PBMCs [31,32]. Similarly, other studies have also reported that T cell proliferation was inhibited by catechin through inhibiting T cell division and cell cycle progression in a dose-dependent manner in vitro [33]. Surprisingly, our results showed that biotransformed catechin metabolites did not induce cell death. The cytotoxic effect of pure catechin and alterations in biotransformed catechin may due to the addition and alteration of hydroxyl groups [34,35]. Moreover, the BCE of biotransformed metabolites showed slight and significant proliferation in unstimulated PBMCs in a dose-dependent manner. This may be due to the presence of high concentrations of various bacterial metabolites, which may act as mitogen for the proliferation of PBMCs. Recent studies have confirmed that polyphenol metabolites are able to penetrate tissue and also permeate into macrophages to be further converted into the methylated form [36].

### 3.3. Effect of Biotransformed Metabolites on Cytokine Secretion in PBMCs

Cytokines are modulators of inflammation and play a key role in acute and chronic inflammation via a complex network of interactions. The influence of catechin biotransformed metabolites on the release of 12 cytokines in unstimulated PBMCs are shown in Figure 4. PBMC incubation with BTMFA increased the production of IL-6 (174.89 pg/mL), IL-8 (68.19 pg/mL), IL-1B (52.55 pg/mL), IL-1A (31.05 pg/mL), IL-10 (30.57 pg/mL) and IL-2 (12.75 pg/mL). However, BTMFB and BCE produced similar quantity of IL-6 (10.36; 10.89 pg/mL) compared to unstimulated control (IL-6, 7.49 pg/mL). The secretion of IL-4, IL-12, IL-17A, IFN-γ, TNF-α, and GM-CSF were suppressed and IL-8 levels were similar in all the tested colonic bacteria-transformed catechin metabolites. In previous studies, epicatechin (EC), epicatechin gallate (ECG), epigallocatechin (EGC) and epigallocatechin gallate (EGCG) were reported to inhibit the production of IL-1A, IL-1B, and IL-6, but increased the levels of IFN-γ, TNF-α, and GM-CSF [37,38]. This obtained data suggests that catechin is transformed into many metabolites by colonic bacteria and these metabolites stimulate human PBMCs differently. Our results indicate that cytokine secretion by PBMCs due to the stimulation of colonic catechin metabolites was dependent on structure, such as monomeric flavanols. The superior effect of BTMFA due to presence of hydroxylated phenolic acid and the length of the side chains of the functional groups present in the metabolites influence the cytokine families [9]. The transformed metabolites that are present in BTMFA such as phenyl propionic acid, benzene carboxylic acid, cinnamic acid and catechol are capable of stimulating human PBMCS to produce inflammatory cytokines [14,39]. The results of IL-1B and IL-6 were in agreement with other studies performed with monomeric and dimeric flavanols of phenolic acids [10]. It was also reported that the simple form of phenolic derivates, such as flavones and flavanols, inhibit TNF-α secretion in macrophages [40]. Phenolic metabolites derived from gut microbial metabolisms modulate the inhibitory or stimulatory activity of PBMCs for cytokine production and may result from transcriptional and post-transcriptional events, which activate a series of cytokines.

### 3.4. Effect of Biotransformed Metabolites on the Expression of IL mRNA in PBMCs

The roles of biotransformed catechin metabolites on the gene expression of cytokines such as IL-4, IL-10, IL-12B, and IL-16 in PBMCs were analyzed (Figure 5). After exposure of BTMFA, BTMFB, and BCE, the gene expression of the target genes were analyzed. Dose-dependent upregulation was observed in IL-4, IL-10 and IL12-B genes of BTMFA-treated PBMCs. BTFA trigger Th2 cells to produce IL-4, IL-6, IL-10 cytokines and thereby regulate the humoral immunity. In other studies catechin and quercetin induce the production and gene expression of Th1 cytokines and down-regulated the Th2 derived cytokines [41]. The expression was 3-fold higher in IL-4, IL-10, and IL-16 in high dose BCE-treated cells due to the higher affinity binding of the receptors in immune cells, thus triggering an intracellular signaling pathway that ultimately regulated the host’s immune response [42]. IL-4, IL-12B, and IL-16 gene expression was downregulated when BTMFB concentration increased. Th2 cells produce IL-4 [43] and play a role in the production of allergen specific IgE, tissue migration of Th2 cells, regulation of tight junctions, and epithelial barrier integrity [44,45]. The downregulation of these genes, which regulate T cells through AhR mediated pathways [46], resulted in the suppression of SP1 protein expression and apoptotic cell death in many cancer cells [47]. Moreover, dihydroxy phenolic metabolites induce the activation of mitogen-activated protein kinase (MAPK) and nuclear factor (NF-kB) pathways, resulting in the activation of dendric cells and regulatory T cells, which contribute to the maintenance of immunotolerance, therefore inhibiting autoimmunity [48]. IL-10 regulates inflammatory interleukin production and antibody response [49]. IL-12B activates the antitumor activity of macrophages and promotes the cytolytic activity of natural killer (NK) cells and lymphokine activated killer cells [50]. IL-16 mediates its biological activity through CD4 and promotes Th1 mediated responses [51]. Th1 cells can inhibit cancer cells by inducing cytotoxic activity in NK cells [52]. NK cells produce perforin and granzyme B which triggers apoptosis and necrosis in cancer cells by creating a hole in the cell membrane, thus facilitating cell destruction [53]. Results clearly suggest that biotransformed catechin metabolites showed immunomodulatory effects in human PBMCs.

## 4. Conclusions

The results in the present study revealed that gut microbes transform catechin by dehydroxylation and C ring cleavage. The biotransformed catechin metabolites stimulate human PBMCs differently. BTMFA and BCE upregulate the production of anti-inflammatory cytokines and downregulate pro-inflammatory cytokines. We conclude that biotransformed catechin metabolites enhance Th2 immune response by anti-inflammatory cytokines, which is more beneficial to overcome inflammatory disorders. Further studies on the use of gut microbe-derived pure catechin metabolites as immunomodulatory agents should be carried out.

## Figures and Tables

**Figure 1 biomolecules-09-00830-f001:**
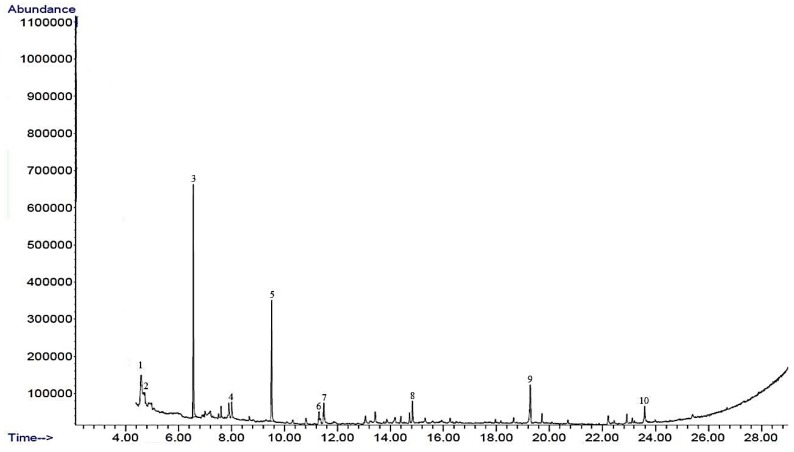
Biotransformed metabolite A (BTMFA).

**Figure 2 biomolecules-09-00830-f002:**
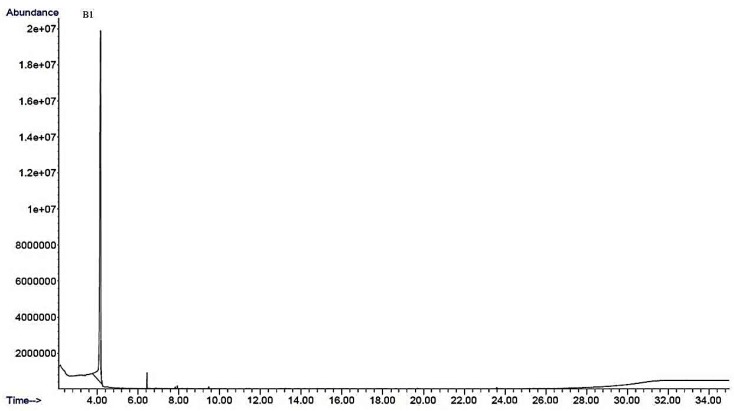
Biotransformed metabolite B (BTMFB).

**Figure 3 biomolecules-09-00830-f003:**
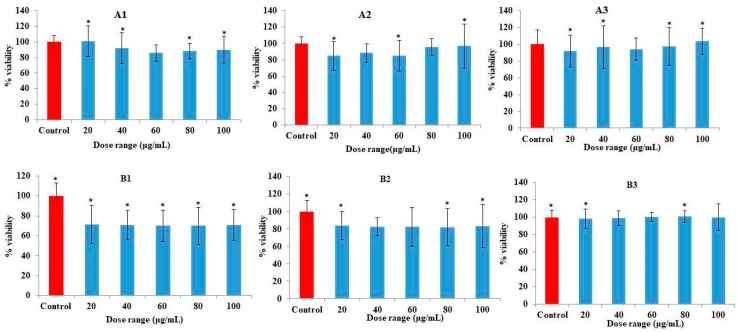
Effects of biotransformed catechin metabolites on the viability of non-stimulated human peripheral blood mononuclear cells (PBMCs) by MTT assay—24 h, (**A**) (**A1**)—BTMFA, (**A2**)—BTMFB, (**A3**)—BCE, and 48 h. (**B**), (**B1**)—BTMFA, (**B2**)—BTMFB, (**B3**)—BCE. Cells were treated with metabolites at concentration of 20, 40, 60, 80, and 100 µg/mL. The columns represent the mean ± standard error of the mean (SEM) of the data from duplicate tests. Asterisk (*) represents data statistically significantly different with respect to the control (*p* < 0.05).

**Figure 4 biomolecules-09-00830-f004:**
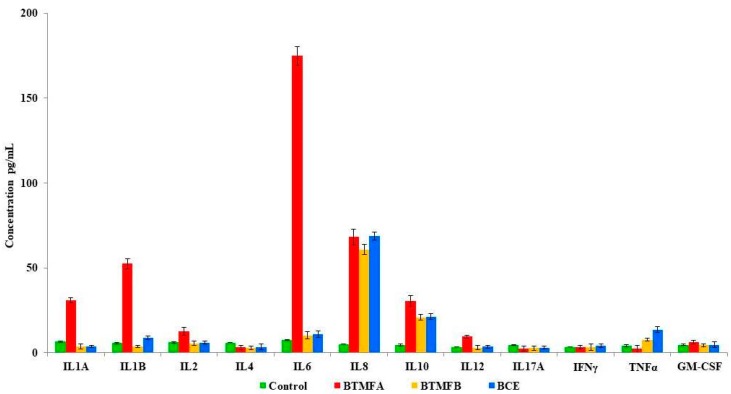
Response of biotransformed catechin metabolites BTMFA, BTMFB, and BCE on release of interleukin (IL)-2, IL-4, IL-5, IL-10, IL-12, IL-13, IL-17A, granulocyte-macrophage colony stimulating factor (GM-CSF), interferon gamma (IFN-γ), transforming growth factor-beta (TGF-β1), and tumor necrosis factor (TNF-α) were measured in PBMCs (duplicates).

**Figure 5 biomolecules-09-00830-f005:**
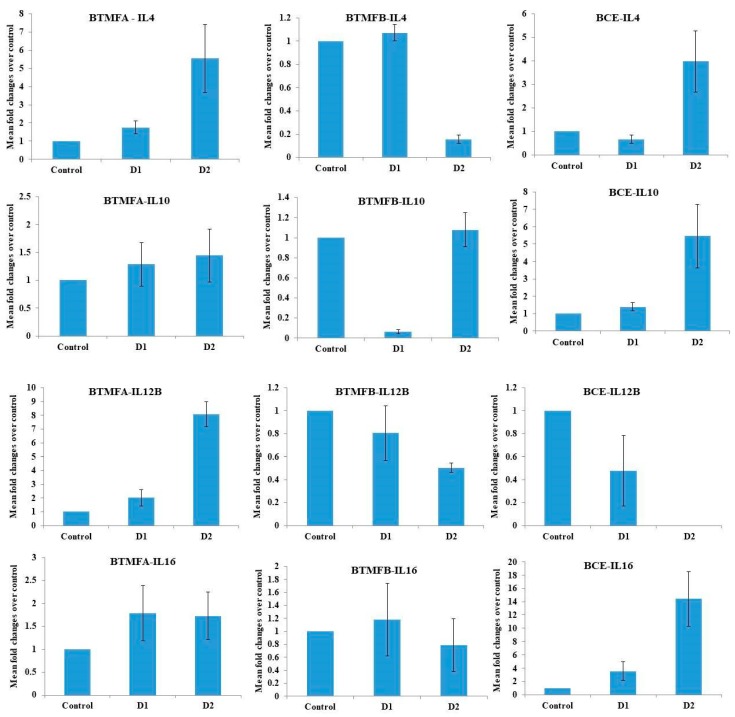
Quantitative reverse transcription-polymerase chain reaction (RT-PCR) analysis of gene expression in human PBMCs. Comparison of change in expression level, shown as fold change after exposure to BTMFA, BTMFB, and BCE for 24 h. Data represent the mean ± SD of three determinations. D1—150 µg/mL, D2—300 µg/mL.

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
