# Peer review of "Colonic Bacteria-Transformed Catechin Metabolite Response to Cytokine Production by Human Peripheral Blood Mononuclear Cells"

_biomolecules, 2019, doi:10.3390/biom9120830_

Round 1
Reviewer 1 Report
Title is unclear.
"effects: should be used instead of "action", and "bacteria" in lieu of "bacterial".
Abstract
*Methods are not properly described
*The authors likely stated Th1 instead of Th2
* The conclusion should be rephrased
* Key words should have catechin and metabolite
* Overall level of English to be improved
Introduction
* "The gut microbiota produces numerous non-toxic metabolites and bioactive molecules which play important roles in the regulation of the immune system." breaks the flow and introduces some confusion.
* Lines 65-68 seems out of place. The research gap addressed by the authors should be clearly stated instead.
*There are multiple typos and grammatical issues that need to be corrected.
Materials and Methods
* Chemicals could be replaced by reagents
* No need to write full names here when they are already in the Introduction
* How many ELISA kits were used?
* Were fecal samples centrifugated at any time? Not clear. If no centrifugation, say so because usually fecal extract preparation involves centrifugation.
* The sentences in Lines 97-99 seem contradictory (pH)
* Ethyl acetate was used as extraction solvent. Are you concerned that you may have missed less polar components?
* Did you perform preparative or analytical TLC?
* Line 131: "90% Acetic acid-water" not clear (what amounts of acetic acid in water?)
Statistical section is too general. The manufacturer for SPSS should be provided. Did you run any post hoc analysis?
* The language level should be improved
Results and Discussion
Lines 191-195 are duplicated in Materials and Methods. I would suggest deleting here.
Lines 246-248: Since your "metabolites" are basically mixtures (at least for BTMFA), it is very difficult to know what's going on since there may be antagonistic effects of actual compounds.
* The authors should explain why they chose different sets of molecules for ELISA and qPCR
* Add limitations
* The language level should be improved
Conclusions
A brief summary of the results is needed.
The current version should be reformulated.
Author Response
Title is unclear.
"Effects: should be used instead of "action", and "bacteria" in lieu of "bacterial".
Response- Thank you for your valuable suggestions, the title of the manuscript is changed “Colonic bacteria-transformed catechin metabolites response to cytokine production by human peripheral blood mononuclear cells”
Abstract
*Methods are not properly described
Response - As per reviewer suggestions, the title of the manuscript is changed accordingly
*The authors likely stated Th1 instead of Th2
Response – Apologize, the typographical error TH1 is removed and TH2 is added in the revised manuscript.
* The conclusion should be rephrased
Response – As per reviewer suggestions, the sentence is rewritten in the revised manuscript.
* Key words should have catechin and metabolite
Response – The key words catechin and metabolites are included in the revised manuscript as per your comments.
* Overall level of English to be improved
Response – Thank you for your valuable comments, Using the help of English editing service the language correction is carried out in the revised manuscript.
Introduction
* "The gut microbiota produces numerous non-toxic metabolites and bioactive molecules which play important roles in the regulation of the immune system." breaks the flow and introduces some confusion.
Response – As per the reviewer suggests, the sentence is removed and rewritten in the revised manuscript.
* Lines 65-68 seems out of place. The research gap addressed by the authors should be clearly stated instead.
Response – Thank you for your valuable suggestions, line 65 – 68 is removed and rewritten carefully to fulfill the research gap.
*There are multiple typos and grammatical issues that need to be corrected.
Response – As per your comments, the typographical errors and grammatical issues are corrected in the revised manuscript.
Materials and Methods
* Chemicals could be replaced by reagents
Response – As per your comments, the word Reagents is added instead of chemicals in the revised manuscript.
* No need to write full names here when they are already in the Introduction
Response – As per your suggestions, the details of interleukin and interferon are removed from the reagents section.
* How many ELISA kits were used?
Response – As per your suggestions, the details of the number of ELISA kit used were included in the revised manuscript.
* Were fecal samples centrifugated at any time? Not clear. If no centrifugation, say so because usually fecal extract preparation involves centrifugation.
Response – Thank you for your suggestion, we have prepared the fecal slurry as per the method described in previous literature and the details were clearly mentioned accordingly.
* The sentences in Lines 97-99 seem contradictory (pH)
Response – Apologize, the typographical error is removed in the revised manuscript.
* Ethyl acetate was used as an extraction solvent. Are you concerned that you may have missed less polar components?
Response – Thank you for your comments, we have focused extraction for biotransformed catechin metabolites based on previous literature. We will consider your suggestions and we will perform in the ongoing research on less polar components.
* Did you perform preparative or analytical TLC?
Response – We have performed the preparative TLC and included in the appropriate place in the revised manuscript as per your comments.
* Line 131: "90% Acetic acid-water" not clear (what amounts of acetic acid in water?)
Response – Thank you for your suggestions, the typographical errors are removed in the revised manuscript.
Statistical section is too general. The manufacturer for SPSS should be provided. Did you run any post hoc analysis?
Response –
* The language level should be improved
Response – Thank you for your valuable comments, language correction is carried out in the revised manuscript.
Results and Discussion
Lines 191-195 are duplicated in Materials and Methods. I would suggest deleting here.
Response – As per you suggestions we have removed the lines 191 – 195 in the revised manuscript.
Lines 246-248: Since your "metabolites" are basically mixtures (at least for BTMFA), it is very difficult to know what's going on since there may be antagonistic effects of actual compounds.
Response – Thank you for the reviewers' valuable comments, In the present preliminary study we aimed to screen the biotransformed catechin metabolites and its response to cytokine by human PBMCs. As per your reviewer's comments, it may act as antagonistic or may not. This point will be rectified from our future study.
* The authors should explain why they chose different sets of molecules for ELISA and qPCR
Response – In the present study, we have analyzed the immune response of biotransformed catechin metabolites. So we selected anti-inflammatory and pro-inflammatory markers in this study.
* Add limitations
Response – Thank you for your comments, in our ongoing research work on catechin biotransformation we are performing some analysis to solve the limitations in this work. We hope we will publish the same in this journal.
* The language level should be improved
Response – Thank you for your valuable comments, language correction is carried out in the revised manuscript.
Conclusions
A brief summary of the results is needed.
Response – As per reviewer suggestions the results included in the revised manuscript.
The current version should be reformulated.
Response – The conclusion part is rewritten in the revised manuscript as per your comments.
Reviewer 2 Report
Review to
Immunomodulatory action of colonic bacterial-transformed catechin metabolites on 1 cytokine production by human peripheral blood mononuclear cells 2
Adisa, Krishnamoorthy, Periyasamy, Athinarayanan, Pandurangan, Alshatwi
The authors wanted to investigate the interesting question of immunological functions of biotransformed catechin metabolites. For this purpose the authors incubated crude fecal bacteria with and without catechin, prepared two fractions of some of these batch-cultures and incubated them with PBMC. They analyzed some preparations in GC/MS. They performed MTT assays and cytokine production assays in PBMC, using the two preparations used for GC/MS. They concluded that metabolites of catechin can induce immune responses.
Although the question is very interesting the study is not sound. Neither the catechin is characterized and tested, nor the hypothetical metabolites, such as B1, 1 to 10 are investigated. The authors cannot conclude that catechin metabolites induce immune responses, since they have not tested the individual metabolites. They can only conclude that the preparation, containing 11 metabolites has some activity. However, the stimulation assay is not well-controlled and the individual metabolites are not investigated. Thus, the reviewer has many major concerns, listed below.
TITLE
The authors state in their title that metabolites of catechin have immunmodulatory actions. This statement is wrong, since they did not test individual metabolites. They can only state that a preparation containing 11 metabolites has some activity.
INTRODUCTION
The authors propose that the investigation of immunmodulatory activities of catechin metabolites is interesting a make this the aim of the study. This indeed would be an interesting question. However, the authors do not address this topic, since they do not investigate metabolites, but investigate only partially purified samples, containing multiple metabolites.
It is unclear to the reviewer why the authors used such a crude system. Why didn´t they use an isolated gut bacterium?
METHODS
In the methods it is unclear which ELISA kits the authors used. Provide a more specific product name, or even better a list of order numbers (compare below line 164).
The source and quality of catechin (line 100) is not described. Since this compound is the basis of the study its source (isolated, recombinant?) and quality (SDS-PAGE, homogeneity) have to be described.
In stimulation assays the culture conditions are very important. In chapter 2.7 and 2.8 the authors do not mention the culture conditions precisely enough. They should always include the cell concentrations, culture volumes, dishes used and incubation times. Is the culture serumfree or does it contain serum? Did the authors perform optimization of culture conditions? Please discuss the latter.
In this context, it is also unclear from how many people/cell isolations and biochemical isolations the BCE, BTMFA and BTMFB are derived. That means, did you prepare pools for BTMFA and BTMFB? Is the BCE also a pool? Compare line 191 to 193 (Results). This procedure and outcome must be described more precisely in the methods
Is the "biotransformed crude extract (BCE)" mentioned at line 147 identical to the "metabolites harvested at 24 h and 48 h" (line 103), or is it "The resulting supernatants" of the centrifugation mentioned at line 107. The term "BCE" must already be defined at this place, otherwise it is unclear what "BCE" is, and this would not be acceptable. It is too late to mention it at line 147.
At line 164 you specify the cytokine ELISA a little more specific (line 164). Please discuss the detection limits of the tested cytokines. Many cytokines are produced in very low amounts. Does the detection limit sufficiently cover these low levels?
RESULTS
Chapter 3.1
In the profiling of the A and B preparations it is unclear how the authors can be sure that the products shown in Fig. 1 are derived from the catechin and not culture products, derived from the bacteria or culture material? Is the product B1 in Fig. 2 similar to any product in Fig. 1? How can such a compound be derived from catechin?
If the authors would apply the suggested compounds 1 to 11 and B1 individually to their GC/MS, would they appear at the same spots, and where would catechin appear?
The discussion text in 214 - 229 is very general and speculative. The connection to the metabolites described in the paper is not clear. Is there any information about enzymes or particular bacterial species involved in the processing?
The legends to Figs. 1 and 2 are very poor, they do not describe the methods sufficiently. This would be important, since the Methods are not very clear. A very important question has to be answered: are the figures derived from the same sources/samples? Please unify the X-axis to the same level. Since the text at line 191 - 195 is actually not a description of results, but rather a description of the methods used, it should be removed from the results and included into the legend.
Chapter 3.2
In Figure 3 it is unclear what belongs to a and what to b, since a and b are not given in the figure. Is 3a derived from BTMFA and 3b from BTMFB, as described in the results? Then, the figure itself is wrong, because there is no a and b, and A1, A2, and so on, do not appear in the text, not even in the text of the figure legend, which is present above the figure.
With respect to your text 245 - 248: the biotransformed material did not "maintain" viability, but rather "did not induce death", please change this.
In the same text you cite a cytotoxic effect of catechin. Did you reproduce this with your catechin? This would be very important and should be included into the paper.
In 248 - 252 you mention the presence of bacterial metabolites. Could these metabolites also appear in the GC/MS presented in Figures 1 and 2? How did you differentiate them from catechin metabolites (compare above - response to 3.2/first paragraph)?
254 - 257 is only speculation and not supported by the data, since you did not test any individual metabolite, you should remove that.
The figure 3 itself it not acceptable. Text and figure do not coincide. It is unclear where a and where b is. The title "BTMFA, BTMFB, BTMBCE" and the legends A1, A2, and so on do not coincide. And to make it even more unclear: is it BCE, BTMBCE or "crude extract" what you investigate. You should use only one determination of a compound throughout the whole paper.
Do you propose the text in 264 - 269 to be result text or methods text or legend text? This text actually does not belong to the results. It has to be included into the legend text, since it is a description of the methods used in the test. However, it makes the confusion even larger between the text in 238 ff., the figure itself and this text. This whole chapter has to be rewritten.
Chapter 3.3
In Figure 4 another determination of the samples is used, and it again does not coincide with the text in 271ff.. Is BTMFA identical to "RA", BTMFB identical to "RB" "Control" identical to BCE or "Crude" identical to BCE? What is "Control"? The determination of the samples has to be unified in the whole paper.
At this place I recalled that you were describing in the Methods that you have also prepared a batch-culture without catechin. However, it was not investigated in 3.1 and 3.2. Is "Control" in Figure 4 the batch-culture without catechin, however, not separated in A and B?
On the other hand, if my guesses are correct and Control means PBMC without stimulation, then you have to check line 276/277. There you state that "and crude extract produced high levels of ... and IL‑6 ... compared to control". However, the control is not lower than the RB and crude levels. This sounds wrong.
If you are studying activation of cytokine production, the inclusion of an unstimulated and a stimulated condition is essential. It has to be included, in order to make obvious how potent a new stimulus is. In your case you should include a measurement of unstimulated cells and LPS-stimulated cells. For activation 100 ng/ml of LPS would be sufficient.
For a sound study, also the testing of the unseparated batches should be included. That means: PBMC +/- catechin, batch with catechin, batch without catechin, A and B of both. Even more important, and essential to answer the aim of the study, would be to test the identified substances separately.
Your conclusion at line 282ff. that "colonic catechin metabolites have different effects on human PBMC" is wrong. You only show that a preparation containing many of the metabolites (i.e., 1 to 10) is stimulating PBMC differently, as compare to a preparation containing only 1 metabolite (i.e. B1).
You also have to make sure that the observed stimulation is not due to bacteria, which are plenty in the fecal preparation, or compounds derived from the bacteria, and might be a part of the preparations you have prepared (e.g., LPS or MDP). A Limulus test for endotoxin may for example exclude that your preparations are contaminated by LPS. Please include such an analysis into the paper.
Please explain, how you can derive the conclusion that stimulation is dependent on "monomeric flavanols" (line 285). Did you test them separately? Are they 95 % of your preparation A? This is only an assumption, may be supported by some literature, but not by your own data. Explain, or remove.
Please discus also that some flavonols are pro- and some are anti-inflammatory. Which are doing what? Which are in your preparation?
Very irritating are the numbers in brackets in the text. There are no standard deviations presented. There are also no SDs in the figure. Does this mean, you performed one measurement for BTMFA and two for BTMFB? This would not be sufficient for publication. Usually three or more measurements are performed. Please explain in detail how many replicates in how many measurements were included in the study presented in Figure 4.
In the legend to Figure 4 the explanation of Control, RA, RB, Crude is missing. Also missing is the explanation of the number of experiments performed. Line 299 - 301 should be a part of the legend.
Chapter 3.4
Yet again, a new determination is introduced, the "CE". Is this identical to BCE? It is standard that one abbreviation is used for one compound throughout a manuscript.
At line 305/306 you state that "fold changes ... at the mRNA level were measured". This is very imprecise and actually not correct. You calculated and presented the changes, however, you measured the mRNA levels of the various cytokine genes.
And probably another misunderstanding. At line 306/307 you state that "dose dependent regulation of BTMFA gene expression ...". However, not the "BTMFA gene" was measured, but rather the cytokine genes. Please rewrite.
In the figure you mention D1 and D2, however, do not explain them in the legend. Again I have to guess, are they two doses? If yes, please explain which.
Again, you probably have one legend above the figure (line 332 and 333 ?), and another below (line 336 ?). This is irritating. If 336 is indeed intended to be legend, it is much to short. D1 and D2 have to be explained. As done in the text in 303 - 306, the samples and measurements should be described. The lines 303 - 306 should then be drastically shortened or removed from the results, since they are to the largest extend methods and/or legend.
In line 316 the authors state that "IL‑6 and IL‑10 gene expression" is upregulated. Unfortunately there is no IL‑6, but only IL‑4, IL‑10, IL‑12 and IL‑16 present in the gene expression Figure 5. They then state that this is "resulting in T-cell proliferation ...". This mixture of data (which are partially not even present) and literature is very irritating. The authors persuade the reader that T-cell proliferation and other lymphocyte functions are caused by the measured cytokines. However, they did not perform any of such measurements. The sentence could read "Our data showed that IL‑10 gene and protein expression was upregulated, which might indicate a role in T-cell proliferation, B-cell proliferation or survival, as discussed previously (Hirano et al., 1985)". !!! Although I am not really convinced that IL‑10 was already described at 1985. !!! Thus, this paper does not really seem to be appropriate.
In line 317 to 329, probably intended as discussion, numerous papers are cited regarding lymphocyte activation. Actually, this does not have any relevance for the submitted paper, since the authors do not show any lymphocyte data, nor did they analyse any "dihydroxy phenolic metabolites", but only show induction of cytokines by crude partially purified materials in leukocyte mixtures. This part might explain in the introduction (if drastically shortened) the rational for measuring IL‑6, IL‑10, IL‑12B or IL‑16, but has no connection to the data of the authors.
In the discussion the reviewer is missing a discussion of the presented data. For example, the authors did not even discuss whether or not the gene expression data (Figure 5) support the protein data (Figure 4). In the reviewer´s opinion this is not the case, thus, not supporting each other, although it should.
The authors should point out more explicitly which data in the paper support each other. At some places the data appear to be in line with each other or the literature, at some other places they do not.
The last sentence of the paragraph (329-331) is unclear and unspecific. Which "transformed phenolic metabolites" the authors refer to? They have not tested any, but only crude samples, containing many substances. The origin is not even clear, is it from the medium, the bacteria or the catechin? Ironically, the authors themself state in this sentence that they are "derived from intestinal bacteria", rather than "processed by" them.
CONCLUSION
The conclusion is unspecific and not based on the data, since the aim of the study, to investigate metabolites was not fulfilled.
REFERENCES
The list of references is very long. Although it might be informative, the reader would certainly get more out of the references if the authors would concentrate on the most relevant papers. It is not necessary to cite up to 6 papers for one point.
Many references used in the results/discussion part are not useful in the discussion, since they have no relation to the actual data, but only make general points (compare above). They should used for introducing the cytokines or cells in the introduction.
-is rechner and kroner 2004 or 2005 (46, 484)
-why is gao (53, 287), rios (225), gonthier (227, 228), hirano (318), engelhardt (322) underlined
-since the field is evolving the authors should not mention "Certain studies" and do not give the relevant papers. Please provide papers for the individual cytokines, or concentrate on those, which are relevant for the paper.
LEGENDS
The legends are very poor, no hint to the methods used, no description of the samples. It is also unclear, how many times the experiments have been performed. This actually must be included.
MINOR COMMMENTS
line 4 - is Alshatwi also affiliation 1?
line 87 - (, .)
line 106 - it should read "7,000 x g"
line 140 - "2500" or "2,500"?
lines 77, 142, 159 - did you use "RPMI-1640" or "RPMI 1630" or "RPMI 1640"? Avoid different abbreviations for the same thing!
Author Response
We would like to appreciate and thank you for your valuable comments to improve the quality of our manuscript. We carefully noted each comment and carried out accordingly in the revised manuscript.
TITLE
The authors state in their title that metabolites of catechin have immunmodulatory actions. This statement is wrong, since they did not test individual metabolites. They can only state that a preparation containing 11 metabolites has some activity.
Response – As per the reviewer suggestion we have modified the title “Colonic bacteria-transformed catechin metabolites response to cytokine production by human peripheral blood mononuclear cells”
INTRODUCTION
The authors propose that the investigation of immunmodulatory activities of catechin metabolites is interesting a make this the aim of the study. This indeed would be an interesting question. However, the authors do not address this topic, since they do not investigate metabolites, but investigate only partially purified samples, containing multiple metabolites.
It is unclear to the reviewer why the authors used such a crude system. Why didn´t they use an isolated gut bacterium?
Response - As per your suggestion, we have mentioned the details of why the crude system is used in the current study. “The crude system of gut microbes produce a larger toolkit of enzymes that catalyze the diverse range of chemical reactions results transformation of catechin. However, this alteration can cause increased activity or prevent them from working on human health. Also, presently we are studying in the isolated gut bacterium”
METHODS
In the methods it is unclear which ELISA kits the authors used. Provide a more specific product name, or even better a list of order numbers (compare below line 164).
Response – As per reviewer comments we have incorporated the details of the ELISA kit in the revised manuscript.
The source and quality of catechin (line 100) is not described. Since this compound is the basis of the study its source (isolated, recombinant?) and quality (SDS-PAGE, homogeneity) have to be described.
Response – we carefully considered the reviewer comments in the revised manuscript and included the details of catechin
In stimulation assays the culture conditions are very important. In chapter 2.7 and 2.8 the authors do not mention the culture conditions precisely enough. They should always include the cell concentrations, culture volumes, dishes used and incubation times. Is the culture serumfree or does it contain serum? Did the authors perform optimization of culture conditions? Please discuss the latter.
Response - As per your kind suggestions we have provided the culture conditions, culture volume, dishes, incubation time and other relevant details in the revised manuscript.
In this context, it is also unclear from how many people/cell isolations and biochemical isolations the BCE, BTMFA and BTMFB are derived. That means, did you prepare pools for BTMFA and BTMFB? Is the BCE also a pool? Compare line 191 to 193 (Results). This procedure and outcome must be described more precisely in the methods
Response – Thank for your valuable comments, we have collected 100 mL of venous blood from single healthy donor for our entire study. We have prepared pools for BTMFA and BTMFB from two triplicates and we clearly mentioned in the revised manuscript.
Is the "biotransformed crude extract (BCE)" mentioned at line 147 identical to the "metabolites harvested at 24 h and 48 h" (line 103), or is it "The resulting supernatants" of the centrifugation mentioned at line 107. The term "BCE" must already be defined at this place, otherwise it is unclear what "BCE" is, and this would not be acceptable. It is too late to mention it at line 147.
Response – We apologize for this. The typographical error has been removed and the sentences have modified accordingly. The term BCE defined at line 107.
At line 164 you specify the cytokine ELISA a little more specific (line 164). Please discuss the detection limits of the tested cytokines. Many cytokines are produced in very low amounts. Does the detection limit sufficiently cover these low levels?
Response- Thank you for your valuable suggestions, as per the Qiagen protocol the antibodies detect sufficiently cover these low levels.
RESULTS
Chapter 3.1
In the profiling of the A and B preparations it is unclear how the authors can be sure that the products shown in Fig. 1 are derived from the catechin and not culture products, derived from the bacteria or culture material? Is the product B1 in Fig. 2 similar to any product in Fig. 1? How can such a compound be derived from catechin?
Response – Thanks for your valuable comment. We have analyzed GC-MS for culture media with bacterial consortia and culture media with catechin. The obtained results confirmed that catechin metabolites corresponding peaks are not exist in the culture media with bacterial consortia and culture media with catechin.
If the authors would apply the suggested compounds 1 to 11 and B1 individually to their GC/MS, would they appear at the same spots, and where would catechin appear?
Response- Thanks for your valuable comments. We have used the mixed consortia for biotransformation, which may transform the catechin completely. So, we did not observe any peaks corresponding to catechin.
The discussion text in 214 - 229 is very general and speculative. The connection to the metabolites described in the paper is not clear. Is there any information about enzymes or particular bacterial species involved in the processing?
Response- Thanks for your valuable comments. We have included the information about bacterial species involved in the catechin transformation in the revised manuscript.
The legends to Figs. 1 and 2 are very poor, they do not describe the methods sufficiently. This would be important, since the Methods are not very clear. A very important question has to be answered: are the figures derived from the same sources/samples? Please unify the X-axis to the same level. Since the text at line 191 - 195 is actually not a description of results, but rather a description of the methods used, it should be removed from the results and included into the legend.
Response – As per reviewer suggestions, the legends have improved as per reviewer suggestions. Figures 1 and 2 derived from BTMFA and BTMFB. We apologize for the GCMS spectra (Figures 1 and 2). The x-axis of obtained chromatogram generated by default mode in the software. As per reviewer suggestions we have improved the legends and removed from the results.
Chapter 3.2
In Figure 3 it is unclear what belongs to a and what to b, since a and b are not given in the figure. Is 3a derived from BTMFA and 3b from BTMFB, as described in the results? Then, the figure itself is wrong, because there is no a and b, and A1, A2, and so on, do not appear in the text, not even in the text of the figure legend, which is present above the figure.
Response - Thank for your valuable suggestions, we clearly mentioned the details of the figure 3a and b in the revised manuscript.
With respect to your text 245 - 248: the biotransformed material did not "maintain" viability, but rather "did not induce death", please change this.
Response - Thank for your valuable suggestions, we have changed the sentence accordingly.
In the same text you cite a cytotoxic effect of catechin. Did you reproduce this with your catechin? This would be very important and should be included into the paper.
Response - We apologized for that and did not reproduce in this manuscript and we will have planned to study in further research.
In 248 - 252 you mention the presence of bacterial metabolites. Could these metabolites also appear in the GC/MS presented in Figures 1 and 2? How did you differentiate them from catechin metabolites (compare above - response to 3.2/first paragraph)?
Response – Thank you for your valuable suggestions, the crude extract (BCE) has many peaks we did not distinct and it may be present. Only the partially purified BTMFA and BTMFB we differentiate the catechin metabolites.
254 - 257 is only speculation and not supported by the data, since you did not test any individual metabolite, you should remove that.
Response – As per your suggestions, we have removed the data in the revised manuscript.
The figure 3 itself it not acceptable. Text and figure do not coincide. It is unclear where a and where b is. The title "BTMFA, BTMFB, BTMBCE" and the legends A1, A2, and so on do not coincide. And to make it even more unclear: is it BCE, BTMBCE or "crude extract" what you investigate. You should use only one determination of a compound throughout the whole paper.
Response – As per your suggestions, figure 3 and the corresponding text have corrected. The figure legends are clearly mentioned in the revised manuscript.
Do you propose the text in 264 - 269 to be result text or methods text or legend text? This text actually does not belong to the results. It has to be included into the legend text, since it is a description of the methods used in the test. However, it makes the confusion even larger between the text in 238 ff., the figure itself and this text. This whole chapter has to be rewritten.
Response – Thank you for your valuable suggestions, the text have included in the legends and rewritten accordingly.
Chapter 3.3
In Figure 4 another determination of the samples is used, and it again does not coincide with the text in 271ff.. Is BTMFA identical to "RA", BTMFB identical to "RB" "Control" identical to BCE or "Crude" identical to BCE? What is "Control"? The determination of the samples has to be unified in the whole paper.
Response – we apologize for the typographical errors. As per the reviewer, we have altered the sentence in the revised manuscript.
At this place I recalled that you were describing in the Methods that you have also prepared a batch-culture without catechin. However, it was not investigated in 3.1 and 3.2. Is "Control" in Figure 4 the batch-culture without catechin, however, not separated in A and B?
Response – Thank you for your valuable suggestions, we focused on the unique TLC spots (BCE, BTMFA, and BTMFB) which did not appear in the untreated control (batch-culture without catechin).
On the other hand, if my guesses are correct and Control means PBMC without stimulation, then you have to check line 276/277. There you state that "and crude extract produced high levels of ... and IL‑6 ... compared to control". However, the control is not lower than the RB and crude levels. This sounds wrong.
Response – Thank you for your valuable suggestions, the line 276 and 277 rewritten accordingly in the revised manuscript.
If you are studying activation of cytokine production, the inclusion of an unstimulated and a stimulated condition is essential. It has to be included, in order to make obvious how potent a new stimulus is. In your case you should include a measurement of unstimulated cells and LPS-stimulated cells. For activation 100 ng/ml of LPS would be sufficient.
Response – Thank you for your valuable suggestions, we designed our present study based on previous literatures, several studies revealed without LPS. However, we will carry out further experimental design with LPS as per your suggestions.
For a sound study, also the testing of the unseparated batches should be included. That means: PBMC +/- catechin, batch with catechin, batch without catechin, A and B of both. Even more important, and essential to answer the aim of the study, would be to test the identified substances separately.
Response – Its very elaborated study, the present study we focused on biotransformed fractions. To delineate elaborately, the study will be carryout in further research.
Your conclusion at line 282ff. that "colonic catechin metabolites have different effects on human PBMC" is wrong. You only show that a preparation containing many of the metabolites (i.e., 1 to 10) is stimulating PBMC differently, as compare to a preparation containing only 1 metabolite (i.e. B1).
Response – As per reviewer suggestions the line 282 is rewritten accordingly in the revised manuscript.
You also have to make sure that the observed stimulation is not due to bacteria, which are plenty in the fecal preparation, or compounds derived from the bacteria, and might be a part of the preparations you have prepared (e.g., LPS or MDP). A Limulus test for endotoxin may for example exclude that your preparations are contaminated by LPS. Please include such an analysis into the paper.
Response – we fractionated only the unique extracellular TLC spots that response to catechin. We will consider your comments we will extend in our further research.
Please explain, how you can derive the conclusion that stimulation is dependent on "monomeric flavanols" (line 285). Did you test them separately? Are they 95 % of your preparation A? This is only an assumption, may be supported by some literature, but not by your own data. Explain, or remove.
Response – As per our GCMS data we get peaks corresponding to molecular weight ranges which relevant to monomeric flavanols. We did not test separately, based on the previous literature we discussed.
Please discus also that some flavonols are pro- and some are anti-inflammatory. Which are doing what? Which are in your preparation?.
Response – as per your suggestions we included some pro and anti-inflammatory flavonols in the revised manuscript
Very irritating are the numbers in brackets in the text. There are no standard deviations presented. There are also no SDs in the figure. Does this mean, you performed one measurement for BTMFA and two for BTMFB? This would not be sufficient for publication. Usually three or more measurements are performed. Please explain in detail how many replicates in how many measurements were included in the study presented in Figure 4.
Response – We apologies for that and we corrected the brackets throughout the manuscript. In this preliminary study, we screened the fractionated metabolites response to cytokine production in duplicates. In further studies, we will be carried out the experiment as per your suggestion.
In the legend to Figure 4 the explanation of Control, RA, RB, Crude is missing. Also missing is the explanation of the number of experiments performed. Line 299 - 301 should be a part of the legend.
Response – There was some typographical error, now it is rectified in figure 4 as well as in the legend as per your suggestions.
Chapter 3.4
Yet again, a new determination is introduced, the "CE". Is this identical to BCE? It is standard that one abbreviation is used for one compound throughout a manuscript.
Response – Thank you, we corrected the abbreviations throughout the manuscript.
At line 305/306 you state that "fold changes ... at the mRNA level were measured". This is very imprecise and actually not correct. You calculated and presented the changes, however, you measured the mRNA levels of the various cytokine genes.
Response – Thank you, we have altered the sentence precisely in the revised manuscript.
And probably another misunderstanding. At line 306/307 you state that "dose dependent regulation of BTMFA gene expression ...". However, not the "BTMFA gene" was measured, but rather the cytokine genes. Please rewrite.
Response – As per your comments we have rewrite the sentence in the revised manuscript
In the figure you mention D1 and D2, however, do not explain them in the legend. Again I have to guess, are they two doses? If yes, please explain which.
Response – As per your comments we have explained the dose details in the legends in the revised manuscript
Again, you probably have one legend above the figure (line 332 and 333 ?), and another below (line 336 ?). This is irritating. If 336 is indeed intended to be legend, it is much to short. D1 and D2 have to be explained. As done in the text in 303 - 306, the samples and measurements should be described. The lines 303 - 306 should then be drastically shortened or removed from the results, since they are to the largest extend methods and/or legend.
Response – As per your comments we have explain the dose details in the legends and 303 and 306 removed and legend is rewritten in the revised manuscript.
In line 316 the authors state that "IL‑6 and IL‑10 gene expression" is upregulated. Unfortunately there is no IL‑6, but only IL‑4, IL‑10, IL‑12 and IL‑16 present in the gene expression Figure 5. They then state that this is "resulting in T-cell proliferation ...". This mixture of data (which are partially not even present) and literature is very irritating. The authors persuade the reader that T-cell proliferation and other lymphocyte functions are caused by the measured cytokines. However, they did not perform any of such measurements. The sentence could read "Our data showed that IL‑10 gene and protein expression was upregulated, which might indicate a role in T-cell proliferation, B-cell proliferation or survival, as discussed previously (Hirano et al., 1985)". !!! Although I am not really convinced that IL‑10 was already described at 1985. !!! Thus, this paper does not really seem to be appropriate.
Response – we corrected all typographical errors. We have removed the sentences and references from the text.
In line 317 to 329, probably intended as discussion, numerous papers are cited regarding lymphocyte activation. Actually, this does not have any relevance for the submitted paper, since the authors do not show any lymphocyte data, nor did they analyse any "dihydroxy phenolic metabolites", but only show induction of cytokines by crude partially purified materials in leukocyte mixtures. This part might explain in the introduction (if drastically shortened) the rational for measuring IL‑6, IL‑10, IL‑12B or IL‑16, but has no connection to the data of the authors.
Response – As per reviewer comments, we have altered the introduction and discussion section in the text.
In the discussion the reviewer is missing a discussion of the presented data. For example, the authors did not even discuss whether or not the gene expression data (Figure 5) support the protein data (Figure 4). In the reviewer´s opinion this is not the case, thus, not supporting each other, although it should.
Response – As per reviewer comments, the present gene expression data with relevant references have been discussed in the revised manuscript.
The authors should point out more explicitly which data in the paper support each other. At some places the data appear to be in line with each other or the literature, at some other places they do not.
Response – As per the reviewer comments, we have corrected accordingly. The revised results and discussion part have been clearly explained in the revised manuscript.
The last sentence of the paragraph (329-331) is unclear and unspecific. Which "transformed phenolic metabolites" the authors refer to? They have not tested any, but only crude samples, containing many substances. The origin is not even clear, is it from the medium, the bacteria or the catechin? Ironically, the authors themself state in this sentence that they are "derived from intestinal bacteria", rather than "processed by" them.
Response – As per reviewer comments we have rewritten the last sentence of the paragraph and explained such as “Our results clearly suggest that biotransformed catechin metabolites showed immunomodulatory effect in human PBMCs”
CONCLUSION
The conclusion is unspecific and not based on the data, since the aim of the study, to investigate metabolites was not fulfilled.
Response – The conclusion part is rewritten based on the obtained data as per reviewer comments in the revised manuscript.
REFERENCES
The list of references is very long. Although it might be informative, the reader would certainly get more out of the references if the authors would concentrate on the most relevant papers. It is not necessary to cite up to 6 papers for one point.
Response – As per your comments, the list of references is reduced and relevant papers are cited in the revised manuscript.
Many references used in the results/discussion part are not useful in the discussion, since they have no relation to the actual data, but only make general points (compare above). They should used for introducing the cytokines or cells in the introduction.
Response – As per your comments, general reference is removed and relevant papers are cited in the revised manuscript.
-is rechner and kroner 2004 or 2005 (46, 484)
-why is gao (53, 287), rios (225), gonthier (227, 228), hirano (318), engelhardt (322) underlined
Response – Apologize, the typographical errors are removed in the revised manuscript
-since the field is evolving the authors should not mention "Certain studies" and do not give the relevant papers. Please provide papers for the individual cytokines, or concentrate on those, which are relevant for the paper.
Response – As per reviewer suggestions, we have included the relevant reference accordingly in the revised manuscript.
LEGENDS
The legends are very poor, no hint to the methods used, no description of the samples. It is also unclear, how many times the experiments have been performed. This actually must be included.
Response – Thank you for your valuable suggestions, all the legends are rewritten accordingly.
MINOR COMMMENTS
line 4 - is Alshatwi also affiliation 1?
Response – Alshatwi affiliation is included in the revised manuscript
line 87 - (, .)
Response – Apologies, the typographical error is removed in the revised manuscript
line 106 - it should read "7,000 x g"
Response – as per reviewer suggestions 7,000 x g" is mentioned accordingly
line 140 - "2500" or "2,500"?
Response – Apologies, the typographical error is removed in the revised manuscript
lines 77, 142, 159 - did you use "RPMI-1640" or "RPMI 1630" or "RPMI 1640"? Avoid different abbreviations for the same thing!
Response – As per reviewer suggestions, the difference in the abbreviations in the line 77, 142 and 159 is corrected in the revised manuscript
Round 2
Reviewer 2 Report
No more comments
Round 3
Reviewer 2 Report
no further comments